# Nursing Students’ Informal Learning of Patient Safety Management Activities

**DOI:** 10.3390/healthcare9121635

**Published:** 2021-11-25

**Authors:** Nam-Yi Kim

**Affiliations:** College of Nursing, Konyang University, Daejeon 35365, Korea; namyi8213@gmail.com; Tel.: +82-42-600-8586

**Keywords:** nursing students, patient safety, learning, safety management, patient care

## Abstract

Nursing students require experience in patient safety management to prevent accidents that compromise patient safety. This study examined the mediating effects of informal learning on nursing students’ patient safety management activities. Responses to questionnaires issued to 136 nursing students in South Korea were analyzed. The independent, mediating, and dependent variables used were nursing competencies, informal learning, and patient safety management activities, respectively. Concept validity and model fitness were confirmed using average variance extracted and composite reliability. Model fitness was confirmed using the goodness-of-fit index, normed fit index, Tucker–Lewis index, comparative fit index, and standardized root mean squared residual. The mediating effect was analyzed using the maximum likelihood method, and statistical significance was assessed through bootstrapping. Informal learning mediated the relationship between nursing competence and patient safety management activities. To improve the implementation of patient safety management activities and increase patient safety competence, learning and teaching of specific patient safety-related knowledge, skills, and attitudes need to be improved. For this, informal learning opportunities (e.g., simulation education and clinical practice) must be increased in the nursing curriculum, and the patient safety education capacity should be increased to maintain continuity and connectivity in clinical practice.

## 1. Introduction

Recently, the specialization and division of labor in medical services, increase in information overload, development of large and complex medical institutions with various departments, and time constraints have increased the possibility of patient safety accidents [1]. In particular, the COVID-19 pandemic has resulted in a large number of patients requiring nursing care, and the shortage of health and medical personnel is adding to the difficulties in performing activities and maintaining environments in the interest of patient safety [2]. If patients’ safety is not protected, their lives could be at risk, which may not only lower the credibility of the medical institution and staff, but also lead to consequences such as the extension of patients’ hospitalization periods, poorer quality of medical care, decreased teamwork among healthcare professionals, and financial losses [1,3]. There is a growing interest in patient safety around the world, and healthcare workers have been working to improve the quality of medical care to ensure patient safety.

Nurses constitute the largest number of professionals in the healthcare industry and play an important role in ensuring patient safety, which is at the forefront of medical care [4]. However, nursing quality differs based on nurses’ clinical experience, skills, competence, education, and professionalism [4,5]. Novice nurses (with less than one year of clinical experience) and advanced beginner nurses (with between one and three years of clinical experience) cannot independently or easily manage nursing tasks [6]. Accordingly, if the nursing competence of novice and advanced beginner nurses is enhanced, patient safety management activities can be implemented with a greater efficiency [7]. Nursing college students, who are prospective nurses, also influence patient safety directly when they engage in patient care during their clinical practice and when they manage patient care practices after graduation [8]. However, although nursing students are caregivers for their patients during their clinical practice and participate in various procedures under the supervision of qualified nurses [9], they still lack proficiency in clinical practice and are inexperienced in dealing with patient safety incidents.

The approaches used to implement efficient solutions for various difficulties in clinical practice differ according to the individual’s nursing competence [7]. Nursing competence is defined as the ability to meet the overall nursing needs effectively; participate in a specific nursing specialty; and apply a combination of skills, knowledge, and judgment [10]. Many international organizations use educational methods to equip nursing students with the knowledge, skills, and attitudes necessary to implement safety management activities properly [11]. They aim to strengthen nursing competence by introducing and integrating curricula and education programs on patient safety. Individual competencies formed through formal education affect informal learning in the actual clinical field [12,13].

Unlike formal learning, informal learning refers to the process of working in the field, engaging in a self-learning process, understanding the context of an organization, and building relationships with its members [12,14]. Formal learning is limited because it must be prompt, it cannot properly reflect the needs of each learner, and the clinical field has insufficient opportunities for practical learning [12]. Conversely, informal learning focuses on acquiring knowledge and methods for performing tasks effectively and adapting to the organization, while participating in introspective interactions through work activities. Clinical practice, which involves workplace learning, is also included in the curriculum of nursing college students; it enables them to learn about the social processes through which individuals actively organize their knowledge based on practical contexts that reflect various relationships between patients, caregivers, and the hospital structures that constitute the clinical field. According to previous studies, the ability to learn informally depends on personal ability [12,13] and affects work, organizational commitment, and organizational socialization [15,16]. As nursing students also need to adapt to the clinical practice and work to perform their tasks, informal learning affects their activities for patient safety management.

Patient safety has been identified as an important issue in the healthcare field, and studies have been conducted on the knowledge and attitudes of nursing college students [17], their confidence in practices [18], their competence in patient safety [8], and the patient safety culture [19]. However, research on informal learning, in which the clinical environment acts as an educational tool for nursing students, is scarce. This study aimed to examine how informal learning, which occurs naturally through clinical practice, affects the relationship between nursing students’ competence and patient safety management activities. Further, it aimed to identify the factors that influence the competence with which nursing students perform patient safety management activities as well as the mediating effect of informal learning, and to provide basic data for the development of educational programs on patient safety management for these students.

## 2. Materials and Methods

### 2.1. Design

Exploratory research was conducted to understand the effects of the competence of nursing college students and their informal learning on patient safety management. The independent, dependent, and mediating variables were nursing ability, patient safety management activity, and informal learning, respectively. Structural equations were used to check the indirect and direct relevance of each variable [20]. A hypothetical model is shown in Figure 1.

### 2.2. Participants and Data Collection

Self-report questionnaires were administered to nursing students with clinical experience. For structural equation modeling, the recommended sample size was n = 10–20 participants per observed variable [21], and the minimum sample size of the maximum likelihood method was typically between 100 and 150 [20]. This study included three factors and 11 observed variables, requiring 110–220 samples.

Data were collected from 1 August to 30 September 2019, using structured self-report questionnaires administered to nursing students at three universities in Daejeon, South Korea. Nursing education in Korea involves participation in clinical practice in the 3rd and 4th year of university. Therefore, the selection criteria for the study subjects were college students who had clinical practice experience as 3rd and 4th year nursing students enrolled in university in 2019. To distribute the questionnaire, a notice was posted on the university bulletin board. All participants who agreed to voluntarily participate in the study were informed of the purpose and method of the study, and their freedom of participation or withdrawal. They were also informed that all information provided would be treated as confidential. Written informed consent was obtained from all the participants. Among the 140 questionnaires distributed, four were not returned; therefore, 136 questionnaires were used in the final analysis (response rate = 97.1%).

### 2.3. Ethics Approval

Before conducting the study, ethics approval was obtained from The Gimcheon University Institutional Review Board (approval number: GU-201904-HRa-02-02-P).

### 2.4. Research Instruments

#### 2.4.1. Participant Characteristics

The general characteristics of the participants were investigated according to their age and grade in school. For patient safety-related characteristics, the presence or absence of safety education in hospitals and schools, and the observation of patient safety accidents during clinical practice were investigated.

#### 2.4.2. Nursing Competence

Nursing competence was assessed using Park and Kim’s nursing competence measurement scale, which comprised 29 items classified into knowledge and skill, professionalism, management and leadership, and relationship and cooperation [22]. Each item was rated on a four-point Likert scale ranging from 1 (“not at all”) to 4 (“strongly agree”). In the original study, Cronbach’s α was 0.92; in this study, it was 0.95.

#### 2.4.3. Informal Learning

This was assessed using Moon and Na’s informal learning scale [12], which comprised 19 items classified into job knowledge acquisition, adjustment to organization and contextual understanding, relationship formation, and personal development skill cultivation. Each item was rated on a five-point Likert scale ranging from 1 (“not at all”) to 5 (“strongly agree”). In Moon and Na’s [12] study, Cronbach’s α was 0.89, and in this study, it was 0.93.

#### 2.4.4. Patient Safety Management

For this assessment, Yoo and Lee’s [9] patient safety management activities scale was used, which comprised 15 items classified into preventive nursing activities, medical information verification, and patient identification [9]. Each item was rated on a five-point Likert scale ranging from 1 (“not at all”) to 5 (“strongly agree”). In Yoo and Lee’s [9] study, Cronbach’s α was 0.89, and in this study, it was 0.91.

### 2.5. Data Analysis

Data were analyzed using SPSS (version 25.0; IBM, Armonk, NY, USA) and AMOS (version 21.0; IBM, Chicago, IL, USA). The subjects’ socio-demographic characteristics were analyzed by frequency, percentage, mean, and standard deviation. The correlations between nursing competence, informal learning, and patient safety management were analyzed using Pearson’s correlation coefficient. Concept validity was confirmed using average variance extracted (AVE) and composite reliability (CR). Model fitness was confirmed using the goodness-of-fit index (GFI), normed fit index (NFI), Tucker–Lewis index (TLI), comparative fit index (CFI), standardized root mean squared residual (SRMR), and root mean squared error of approximation (RMSEA). The mediating effect of informal learning on the relationship between nursing competence and patient safety management was identified through a covariate structure analysis using the maximum likelihood (ML) method, and the statistical significance was determined through bootstrapping. The significance level was set at *p* < 0.05.

## 3. Results

### 3.1. General Characteristics and Descriptive Statistics of Participants

The participants’ mean age was 22.10 ± 1.97 years. The mean of the measured variables was as follows: nursing competence 3.23 ± 0.36 (out of 4), informal learning 3.97 ± 0.049 (out of 5), and patient safety management activities 4.18 ± 0.48 (out of 5). Skewness and kurtosis were checked to verify sample normality. Skewness ranged from 0.08 to 0.41 and kurtosis from 0.01 to 0.31. As the skewness values were lower than 2 and kurtosis values were less than 7, the normal distribution of the measurement variables was validated (Table 1) [21].

### 3.2. Correlational and Validity Analysis

The correlations among all pairs of measurement variables were significant. The absolute values of the correlation coefficients ranged from 0.60 to 0.66, and all were below 0.80, thereby confirming that multicollinearity was not a concern. If the AVE is greater than 0.05, and CR is greater than 0.07, concept validity is confirmed. In this study, AVE was 0.63–0.67, and CR was 0.86–0.88 (Table 2).

### 3.3. Fitness of the Hypothetical Model and Path Analysis

The index values for testing model fitness were as follows: χ^2^/DF = 1.73, GFI = 0.91, NFI = 0.93, TLI = 0.96, CFI = 0.97, SRMR = 0.05, and RMSEA = 0.06. The chi-squared test rejected the null hypothesis. However, because the chi-square values are sensitive to the sample size, even the slightest difference between the sample and fitted matrices appeared to reject the model [20]. Other goodness-of-fit indices met the criteria (GFI ≥ 0.90, NFI ≥ 0.90, TLI ≥ 0.90, CFI ≥ 0.90, SRMR ≤ 0.08, and RMSEA ≤ 0.08) [20].

The results for the direct and indirect relationships in the hypothetical model are presented in Table 3. The direct relationship of nursing competence (γ = 0.73, *p* = 0.005) on informal learning was significant, with an explanatory power of 53%. The impact of nursing competence on patient safety management activities was not significant in the direct relationship (γ = 0.26, *p* = 0.114), but was significant in the indirect relationship (γ = 0.43, *p* = 0.005). The direct relationship between informal learning and patient safety management activities was significant (γ = 0.59, *p* = 0.009). The explanatory power of nursing competence and informal learning, with respect to patient safety management activities, was 63% (Table 3).

The direct relationship between nursing competence and patient safety management activities was not significant; however, the indirect relationship was significant, indicating that informal learning fully mediates the relationship between nursing competence and patient safety management activities (Figure 2).

## 4. Discussion

This study evaluated the mediating effect of informal learning on the relationship between nursing competence and the patient safety management activities of nursing students. The ability of nursing college students to perform such activities cannot be improved by enhancing nursing competence alone; informal learning, which is also known as empirical learning from clinical practice in the hospital, is also required. Patient safety management activities are not improved by knowledge and skills; they are improved by applying what is learned empirically through interactions with patients in the clinical field. Informal learning in these real-world situations occurs in practical contexts that reflect various relationships, patients, caregivers, and hospital structures that make up the clinical field.

A previous study reported that implementing patient safety management activities was affected by patient safety competence [23]. The findings of this study indicate that, contrarily, nursing competence does not directly affect such activities, which suggests a difference between competence in nursing and patient safety management. Nursing competence is a comprehensive concept regarding the ability to integrate scientific, ethical, personal, and esthetic knowledge while providing care [22]. Conversely, patient safety competence refers to the knowledge, skills, and attitudes that healthcare providers require to prevent medical errors and provide safe healthcare to patients; it has a more highly specific purpose than nursing competence [24]. To enhance the efficiency of implementing patient safety management activities, it is necessary to provide a specific and practical capacity for patient safety, in addition to the fundamental nursing capacity. According to the American National Institute of Medicine’s education report, medical professionals are not adequately prepared to protect patient safety and improve the quality of care; therefore, professionals require educational training to develop appropriate patient safety capabilities [25].

To strengthen patient safety capacity, specific and practical education regarding patient safety is needed. Informal learning enables students to learn through routine activities that take place during clinical practice. Informal learning fuses various factors such as organizational adaptation and context understanding, relationship formation, and self-development skills, in addition to acquiring work-related knowledge [13]. Further, previous research has reported that informal learning affects organizational socialization [16] and organizational immersion [15], compared to educator-centered formal learning. Therefore, for nursing college students to improve their implementation of patient safety management activities, a greater emphasis on clinical practice in the university’s curriculum is required. However, as the emphasis on patient safety and privacy has increased, it has become difficult for nursing students to provide direct nursing, such as injection or treatment (Foley catheterization, wound disinfection, etc.) [26]. In contrast, the clinical practice of observing patients or viewing electronic medical records is increasing [26]. As patient safety management activities increase with greater work experience [27], it is necessary to increase the clinical practice of nursing college students while ensuring patient safety.

One effective informal learning solution to this problem is to use simulation training and thus enable students to create a clinical site-like space to gain learning experiences in a safe and systematic environment [28]. Simulation is a form of education for nursing college students who lack the opportunity to perform actual nursing care that can detect and correct behavior that threatens patient safety; it also improves skills associated with patient safety competence, such as critical thinking and problem-solving processes, decision-making skills, and clinical reasoning [29,30]. If various simulation programs related to patient safety are developed and used for education purposes, the efficiency of the patient safety management activities of nursing college students may improve.

However, the capacity of university professors of nursing for patient safety education must be supported to provide nursing students with practical simulation education on patient safety. The absence of an instructor with sufficient competence in patient safety can act as a barrier in providing patient safety education for students [31]. A number of previous studies identified the lack of sufficient competent instructors [32], lack of instructors’ self-confidence in patient safety education [33], and differences in clinical competence in teaching and practice [34]. University professors and instructors of nursing need to be trained in the provision of patient safety education and must expand the training opportunities provided to nursing students [31]. In addition, nursing colleges, clinical practice institutions, nursing college professors, and clinical field educators need cooperation to ensure the appropriate continuity and linkage between theoretical knowledge and clinical practice in the nursing curriculum. The role of clinical practice in the nursing college curriculum should be extended, and simulation training should be actively utilized to achieve this goal. Nursing managers in clinical practice hospitals should recognize nursing students as future nurses and support them to ensure that informal learning is fully achieved. Nursing college professors need to improve their patient safety education capabilities to maintain continuity and connectivity in clinical practice.

## 5. Limitations

This study was conducted with nursing students from a few regions. Therefore, future research should include nursing students from various regions. In addition to nursing competence and informal learning, other factors (such as organizational factors) may affect patient safety management activities [34]. These additional factors should be considered in future studies. In addition, since the research was conducted using self-reported subjective data, casual effects between variables could not be confirmed. Experimental research can be used to overcome the limitations of self-reporting in data collection and to obtain objective data. For example, changes in patient safety management activities can be identified in response to the type (simulation education or clinical practice) or the duration of informal learning.

## 6. Conclusions

To improve the implementation of patient safety management activities and increase patient safety competence, the learning and teaching of specific knowledge, skills, and attitudes related to patient safety need to be improved. Particularly, measures to increase informal learning for nursing students should be implemented. For example, it is necessary to provide various opportunities for simulation education in the nursing curriculum or increase nursing opportunities directly in the clinical practice. To this end, nursing professors should develop their capacity for patient safety education, and practice leaders in clinical settings should recognize the necessity of informal learning and support nursing students by providing many opportunities for them to interact with patients.

## Figures and Tables

**Figure 1 healthcare-09-01635-f001:**
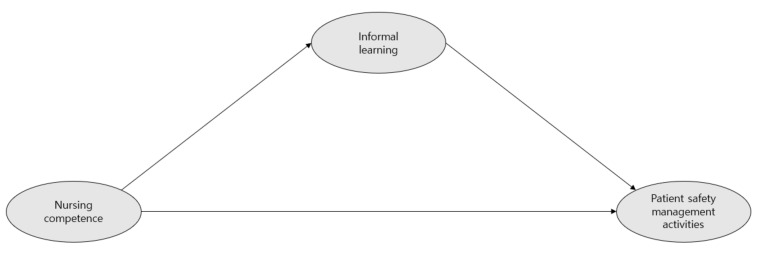
Hypothetical model.

**Figure 2 healthcare-09-01635-f002:**
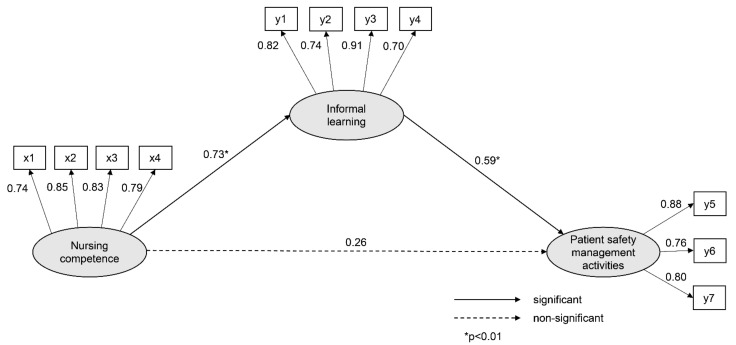
Path diagram of the model. Note: x1: knowledge and skill; x2: professionalism; x3: management and leadership; x4: relationship and cooperation; y1: job knowledge acquisition; y2: organizational adaptation; y3: relationship formation; y4: self-development; y5: preventive nursing activities; y6: medical information verification; y7: patient identification.

**Table 1 healthcare-09-01635-t001:** General characteristics and descriptive statistics of observed variables (*N* = 136).

Variable	Category	*n* (%)	Mean ± SD	Skewness	Kurtosis
Age (years)			22.10 ± 1.97		
University grade	3rd year	67 (49.3)			
	4th year	69 (50.7)			
Safety education (hospital)	Yes	111 (81.6)	1.46 ± 1.78		
	No	25 (18.4)			
Safety education (college)	Yes	95 (69.9)	0.96 ± 1.18		
	No	41 (30.1)			
Observation of patient safety accident	Yes	77 (56.6)	0.72 ± 1.42		
	No	59 (43.4)			
Nursing competence			3.23 ± 0.36	0.41	−0.31
Informal learning			3.97 ± 0.49	−0.08	−0.01
Patient safety management activities			4.18 ± 0.48	−0.08	−0.21

SD = standard deviation.

**Table 2 healthcare-09-01635-t002:** Correlation relations between variables and verification of construct validity.

Variable	NC	IL	PSMA	AVE	CR
	R	r	r		
	(*p*)	(*p*)	(*p*)		
Nursing competence	1			0.65	0.88

Informal learning	0.64	1		0.63	0.87
	(<0.001)				
Patient safety management activities	0.60	0.66	1	0.67	0.86
	(<0.001)	(<0.001)			

Note: NC = nursing competence; IL = informal learning; PSMA = patient safety management activities; AVE = average variance extracted; CR = composite reliability.

**Table 3 healthcare-09-01635-t003:** Verification of the hypothetical model.

Endogenous Variable	Exogenous Variable	SE	CR	*P*	SRW	SMC	Direct Β (*p*)	Indirect Β (*p*)
IL	NC	1.08	7.12		0.73	0.53	0.73 (0.005)	
PSMA	NC	0.37	2.27	0.023	0.26	0.63	0.26 (0.114)	0.43 (0.005)
	IL	0.57	5.00	<0.001	0.59		0.59 (0.009)	
Goodness-of-fit statistics		χ^2^/DF(*p*) = 1.73(.003), GFI = 0.91, NFI = 0.93, TLI = 0.97, CFI = 0.97, SRMR = 0.05, RMSEA = 0.06						

Note: SE = standard error; CR = composite reliability; SRW = standard regression weights; SMC = squared multiple correlation; IL = informal learning; NC = nursing competence; PSMA = patient safety management activities; DF = degrees of freedom; GFI = goodness-of-fit index; NFI = normed fit index; TLI = Tucker–Lewis index; CFI = comparative fit index; SRMR = standardized root mean squared residual.

## Data Availability

Data cannot be shared publicly because of restrictions by the Gimcheon University Institutional Review Board. Data are available from the Gimcheon University Institutional Data Access/Ethics Committee for researchers who meet the criteria for access to confidential data. Data requests can be addressed to the Gimcheon University Institutional Review Board (82-54-420-4427, hyeon62@gimcheon.ac.kr).

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
