# Peer review of "Nursing Students’ Informal Learning of Patient Safety Management Activities"

_healthcare, 2021, doi:10.3390/healthcare9121635_

Round 1

Reviewer 1 Report

Dear Authors

Thank you for the opportunity to review your article.

Brief summary: This cross-sectional and exploratory study aimed to examine the mediating effects of informal learning on nursing students’ patient safety management activities. The findings show that Informal learning mediated the relationship between nursing competence and patient safety management activities.

 Areas of strength

The references included are relevant for the subject under study. But the manuscript only shows 11/33(33%) references from the last 5 years. There is strong concordance between the and the methods used. The discussion correlates with the presented data and takes the published literature into account. The manuscript presents some practical implications.

Topics to improve

Page 2, line 34-35 – please correct. Remove the space between “medical” and “institution and”

Page 7, line 280 - Please write down the limitations of the study.

Author Response

Thank you very much for reviewing my manuscript. The revised manuscript is attached based on the review opinion. The response to you is in the attachment.
Thank you again.

Reviewer 2 Report

Firstly, I thank Editorial Committee for the opportunity to review this manuscript. The author deal with an important issue for undergraduate nursing students related to their informal learning of patient safety management activities. Furthermore, the proposed manuscript meets adequately the purposes of the journal. Although the study includes a large sample and the methodological rigor is adequate, below I suggest some recommendations and comments to improve the quality of the manuscript:

  • Introduction:
    • The Introduction section is well carried out and updated according to the latest studies of this research field.
    • Please, reference the first sentence of the manuscript (line 27 – 39) to support it.
  • Material and Methods:
    • Were other hypothesized mediational models tested? (line 97).
    • Please, clarify nursing study plans in Korea and the year that students were taking. In this sense, what is the meaning of ‘junior’ and ‘senior’ (variable grade in the Results section)?
    • Please, include the response rate in line 115 or the Results section.
    • Please, include the response Likert-scales of each research instrument showing the first and last response option. In this sense, I consider it is necessary to clarify each instrument consist of different subscales or dimensions.
    • Please, correct the reference of Moon and Na’s informal learning scale (the correct reference is [11]).
    • In this section, I consider it is necessary to show and specify participants’ characteristics shown in Table 1 (variables grade, safety education (hospital), safety education (college), and observation of patient safety accident), specifying the measure of these characteristics. In addition, why the university center was not included as a participant characteristic?
  • Results:
    • The variables grade, safety education (hospital), safety education (college), and observation of patient safety accidents are not adequately described. In this sense, did these characteristics impact nursing competence, informal learning, and/or patient safety management? Please, clarify whether differences in the main variables of the study were found based on participants’ characteristics.
    • Please, include a reference to support the interpretation of skewness and kurtosis values (line 155 – 157).
    • In addition to the standardized residual mean root (SRMR) and the comparative fit index (CFI), the root mean square error of approximation (RMSEA) is a widely used index to test hypothesized mediational models. Please include RMSEA and its interpretation supported by the appropriate reference.
    • Please, specify that the obtained mediation was total and not partial (line 196 – 197).
  • Discussion:
    • Please, include more study limitations, mainly related to the design used.
    • Please, include more recommendations for future research.
  • References:
    • Although 36% of references are from 5 years ago and 70% are from 10 years ago, I consider the references are updated regarding the scarce research in this field.

Author Response

(The authors gave the same response as above.)

Round 2

Reviewer 2 Report

Firstly, I thank Editorial Committee for the opportunity to review this manuscript again. I congratulate the author for adequately following all my recommendations and comments to improve the manuscript's quality.